# Joint-Sparing Resection around the Knee for Osteosarcoma: Long-Term Outcomes of Biologic Reconstruction with Vascularized Fibula Graft Combined with Massive Allograft

**DOI:** 10.3390/cancers16091672

**Published:** 2024-04-26

**Authors:** Roberto Scanferla, Federico Scolari, Francesco Muratori, Angela Tamburini, Luca Delcroix, Guido Scoccianti, Giovanni Beltrami, Marco Innocenti, Domenico Andrea Campanacci

**Affiliations:** 1Department of Orthopaedic Oncology and Reconstructive Surgery, Careggi University Hospital, Largo Palagi 1, 50139 Florence, Italy; federico.scolari@unifi.it (F.S.); muratorif@aou-careggi.toscana.it (F.M.); scocciantig@aou-careggi.toscana.it (G.S.); domenicoandrea.campanacci@unifi.it (D.A.C.); 2Department of Paediatric Oncology, Meyer University Hospital, Viale Gaetano Pieraccini 24, 50139 Florence, Italy; angela.tamburini@meyer.it; 3Department of Plastic Surgery, Careggi University Hospital, Largo Palagi 1, 50139 Florence, Italy; delcr@iol.it; 4Department of Paediatric Orthopaedics, Meyer University Hospital, Viale Gaetano Pieraccini 24, 50139 Florence, Italy; giovanni.beltrami@unifi.it; 5Department of Plastic Surgery, Rizzoli Orthopaedic Insitute, Via Giulio Cesare Pupilli 1, 40136 Bologna, Italy; marco.innocenti@ior.it

**Keywords:** osteosarcoma, joint-sparing resection, intercalary, knee, femur, tibia, vascularized fibula

## Abstract

**Simple Summary:**

Osteosarcoma most frequently affects the metaphyseal region of the distal femur and proximal tibia; in around 20% of patients, the epiphyseal bone is not affected and an intercalary joint-sparing resection can be safely performed, preserving the native joint and ligament insertions. In young patients, according to their high functional expectations and potential long-life expectancy, the objective of reconstruction is to restore lower limb function with a low risk of reoperation and implant removal at long-term follow-up. VFG combined with massive allograft is one of the possible reconstructive options after intercalary resection around the knee. In the present study, we aimed to investigate the long-term results of this technique in patients treated for osteosarcoma around the knee with a joint-sparing resection.

**Abstract:**

(1) Background: We aim to address the following questions. What was the complication rate of vascularized fibula graft (VFG) combined with massive allograft in patients treated with joint-sparing resection around the knee for a high-grade osteosarcoma? What was the long-term survivorship of VFG free from revision and graft removal? What were the functional results as assessed by the Musculoskeletal Tumor Society (MSTS) score? (2) Methods: 39 patients treated in our unit for osteosarcoma around the knee with intercalary resection and reconstruction with VFG combined with massive allograft were included; 26 patients underwent intercalary tibial resection, while 13 underwent intercalary femoral resection. (3) Results: Mean Follow-Up was 205 months (28 to 424). Complications that required surgery were assessed in requiring surgical revision in 19 patients (49%) after a mean of 31 months (0 to 107), while VFG removal was necessary in three patients (8%). The revision-free survival of the reconstructions was 59% at 5 years and 50% at 10 to 30 years. The overall survival of the reconstructions was 95% at 5 to 15 years and 89% at 20 to 30 years. The mean MSTS score was 29.3 (23 to 30). (4) Conclusions: VFG represents an effective reconstructive option after joint-sparing intercalary resection around the knee for osteosarcoma.

## 1. Introduction

Osteosarcoma is the most common primary malignant bone tumor in children and young patients and most frequently affects the distal femur and proximal tibia [1]. During the last decades, limb salvage has become the standard surgical option due to the introduction of effective chemotherapy protocols, with an improved patient survival that nowadays is greater than 70% at 5 years [2]. Therefore, a long-term durable reconstruction with good functional outcomes is advised for these patients. A trans-physeal spread of the tumor in metaphyseal osteosarcomas has been observed in approximately 80% of the cases [3], requiring an intra-articular resection to obtain wide surgical margins. Less frequently, when the epiphyseal bone is not affected, through accurate imaging-based planning and the use of intraoperative guides or navigation systems, an intercalary joint-sparing resection can be safely performed [4,5,6], preserving the native joint and ligament insertions.

Several options to reconstruct large intercalary defects of the femur and tibia after resection for osteosarcomas have been described such as intercalary prosthesis [7,8], massive allografts [6,7], bone transport [9], osteoinductive membrane technique [10], recycled autografts [11,12], and vascularized fibula combined or not with massive allografts [13]. 

Reconstruction with vascularized fibula graft (VFG) was first reported in 1975 by Taylor et al. [14] and, two years later, Weiland et al. [15] first described the application in tumor resection. Intercalary VFG provides many advantages such as early osteotomy union, the capacity to hypertrophy under mechanical stress, and spontaneous healing potential after fracture, even in critical soft tissue conditions and in patients treated with chemotherapy [16]. Capanna et al. [13], in 1993, described a reconstructive option after intercalary resection for primary bone tumors in long bones based on the combination of VFG and massive allograft, in order to merge the primary mechanical stability of the allograft with the long-term biological potential of the vascularized fibula. Several authors have reported their results using different reconstructive options to reconstruct large intercalary defects in the lower limb after resection for osteosarcoma [6,12,17]. However, to our knowledge, the literature lacks long-term results of VFG combined with massive allograft reconstruction in this group of patients.

We therefore asked the following questions. (1) What was the complication rate of VFG combined with massive allograft in patients treated with joint-sparing resection around the knee for a high-grade osteosarcoma? (2) What was the long-term survivorship of VFG free from revision and graft removal? (3) What were the functional results as assessed by the Musculoskeletal Tumor Society (MSTS) score?

## 2. Materials and Methods

### 2.1. Patients

All patients surgically treated between 1988 and 2021 for osteosarcoma around the knee with intercalary resection of the femur or the tibia and reconstruction with a VFG combined with massive allograft were reviewed. Exclusion criteria were intercalary resections in the lower limb with the osteotomy around the knee farther than 6 cm from the knee joint, reconstruction with massive allograft or VFG alone, and less than 24 months follow-up. We usually employ reconstruction with VFG and massive allograft after intercalary resection of the femur and tibia in young or adult patients treated for localized osteosarcoma with good predicted life expectancy and high functional demands, in whom at least 1 cm of residual bone stock of the epiphysis around the knee can be preserved. In this group of patients, we routinely prefer this reconstructive technique after resection longer than 10 cm, while in shorter resection, we usually employ intercalary massive allografts alone; in diaphyseal resections in children, a VFG alone is usually adopted. On the contrary, we avoid VFG in patients older than 70 years, in tumoral progression under chemotherapy, or in metastatic patients with poor prognosis; in these clinical situations, alternative reconstructive options are usually adopted such as massive allografts, intercalary prosthesis, plate and cement, or intramedullary nail with cement augmentation. According to the above-mentioned indications, thirty-nine patients were included in this study and their characteristics are summarized in Table 1.

Overall, 26 patients underwent intercalary tibial resection, while 13 underwent intercalary femoral resection. In all patients, VFG was used in combination with massive allograft. 

The mean age of the patients at the time of surgery was 17 years (5 to 67), 29 of them were skeletally immature, while 10 reached skeletal maturity. All the patients had a diagnosis of osteosarcoma, high-grade in 34 (osteoblastic in 25, fibroblastic in 5, and chondroblastic and teleangectatic in 2), and low-grade in 5 (parosteal in 3 and fibroblastic in 2). At diagnosis, four patients had a pathologic fracture and, according to the MSTS staging system [18], one tumor was Stage IA, four IB, one IIA, and 33 IIB. All patients with a diagnosis of high-grade osteosarcoma received pre- and post-operative chemotherapy, except for one 67-year-old patient who had only postoperative treatment; in these patients, the surgery was usually planned 3 weeks after the last drug administration, not differently from other reconstructive options. No patients underwent radiation therapy.

All subjects gave their informed consent for inclusion before they participated in the study. The study was conducted in accordance with the Declaration of Helsinki and the protocol was approved by the Ethics Committee of Area Vasta Centro Toscana (ref. 10197/2017)

### 2.2. Surgical Technique, Aftercare, and Cancer Treatment

All operations were performed with the patient in the supine position, through an extended lateral approach for femoral resections and through an anteromedial or anterolateral approach for tibial resections, according to the prevalent tumor extension. In 14 patients, an intra-epiphyseal resection was performed (3 femoral and 11 tibial). The mean resection length was 14.5 cm (10 to 25) and the mean femoral resection length was 15 cm (10 to 25), while the mean tibial resection length was 14 cm (10 to 20). Considering femoral tumors, the intercalary resections left a mean residual distal juxta-articular bone of 3.8 cm (2 to 6) and a proximal residual bone of 16.5 cm (6 to 26); while considering tibial tumors, the resections left a mean residual proximal juxta-articular bone of 3 cm (1 to 6) and a distal residual bone of 16 cm (7 to 21). In terms of the histologic examination of the resected tumors, surgical margins were wide in all patients, except for one in which they were marginal. Tumor necrosis on resected specimens was ≥90% in sixteen patients and <90% in seventeen patients.

The vascularized fibula was harvested from the contralateral leg in all femoral resections and in 23 of the 26 tibial resections; an ipsilateral pedicle fibula was used in three patients who underwent tibial resection. The free VFG was harvested by a microsurgical team using separate instruments, taking care to avoid contamination between the two surgical fields, through a posterolateral approach. The pedicle VFG was harvested through a posterolateral approach, with a double incision on the same leg. The mean length of the fibular graft was 17 cm (12 to 28). The harvested fibula was at least 2 cm longer than the intercalary resection length to obtain at least 1 cm overlap for each osteotomy. Primary syndesmotic screw fixation was performed at the ankle of the donor side only in one child; we routinely do not use primary syndesmotic fixation in patients with more than 7 cm of residual distal fibula. In patients treated with free VFG reconstruction, the vascular pedicle of the flap, including the peroneal artery and two vena comitans, was anastomosed with the collateral branch of the superficial femoral vessels in patients with femoral reconstruction and with the collateral branch of anterior tibial vessels in patients with tibial reconstruction. Furthermore, in three patients with tibial reconstruction, the free VFG was harvested with the fascio-cutaneous flap based on its perforator vessels, in order to cover the skin defect secondary to tumor excision.

During the fibular harvesting, the massive allograft was thawed in a warm antibiotic solution, sectioned to the proper length, and shaped to receive the fibular graft. Bone fixation was performed with a bridging plate in 19 patients, metaphyseal screws and diaphyseal plate in 13 patients, only screws in 4 patients, two plates (one proximal and one distal) in 2 patients, and double plate (one bridging and two separate at osteotomies) in 1 patient. A single or double bridging plate was classified as a bridging fixation (Figure 1); metaphyseal screws and a diaphyseal plate, only screws or two separate plates were considered as non-bridging fixation (Figure 2). Ten patients underwent bridging fixation either for tibial (38%) or femoral (77%) reconstructions. The mean operative time was 7.5 h, ranging between 4.5 and 10 h.

All patients received perioperative antibiotic prophylaxis following the protocol of our Institution, which was continued until drain removal. After surgery, in tibial reconstructions in which the patellar tendon was reattached and in pediatric age, the operated limb was protected with a long leg cast for four weeks. Controlled passive movements of the joints were then allowed. On the contrary, in young and adult patients with intercalary femoral resections or tibial resections distal to the tibial apophysis, the passive movements of lower limb joints were immediately allowed. No weight-bearing was allowed until radiographic evidence of VFG union; then, partial weight-bearing with crutches was started. Full weight-bearing was eventually granted after evidence of complete allograft union. The donor’s leg was left free after surgery, encouraging active and passive movements of the knee, ankle, and toes. Full weight-bearing on the donor side was then allowed after 3 weeks from surgery.

### 2.3. Data Sources and Variables

All patients were periodically reviewed according to oncologic follow-up, undergoing clinical and radiologic examinations. After surgery, we reviewed patients with malignancy every 3 months during the first 2 years, every 4 months during the third year, and every 6 months until the tenth year after primary surgery. Data extracted from medical records were registered in our database.

The mean follow-up of the patients included in this study was 205 months (28 to 424) and, although five patients have not been seen in the last 5 years and were not known to have died; they had 116, 137, 165, 262, and 265 months of follow-up, respectively, and were included since they had the minimum required follow-up of 24 months. At the last clinical follow-up, 25 of 39 patients (64%) were continuously disease-free. Six patients had no evidence of disease after treatment of local recurrence in two cases and distant metastasis in four. Seven patients died of disease after a mean of 76 months (28 to 128) from index surgery due to metastatic disease, while one patient died from another cause after 36 months. Overall, patient survival was 95% (95% CI 88% to 100%) at 5 years, 86% (95% CI 75% to 98%) at 10 years, and 82% (95% CI 70% to 96%) at 15 years to 30 years.

The functional results were assessed using the Musculoskeletal Tumor Society (MSTS) score [19], which is a well-known system to evaluate functional outcomes in patients treated for bone tumors; in the lower limb reconstructions, it evaluates six parameters: pain, function, emotional acceptance, supports, walking, and gait, giving a value ranging from 0 to 5. The sum of the individual scores defines the overall functional score with a maximum of 30 points [19]. The MSTS score was evaluated in patients in whom the graft was retained at the last clinical control. In all patients, the union of the VFG and allograft was radiographically assessed and complications and graft removal were registered during follow-up. At the same time, donor site morbidity was evaluated. We evaluated graft union and hypertrophy as signs of persistent vascular supply, while we did not routinely use any method of monitoring fibular vitality, such as VFG with skin flap based on perforator vessels, bone scan, or single-photon emission computed tomography (SPECT). We defined union on X-rays as a cortical fusion of allograft and of VFG on anterior–posterior and lateral views, while we defined nonunion as the absence of osteotomy union on radiographs 9 months after the index surgery, with or without loosening of fixation.

### 2.4. Statistical Analysis

Survival of the reconstruction was determined according to the Kaplan–Meier method, including revision surgery for any complications and removal of the VFG as endpoints indicating failure. Kaplan–Meier curves and survival proportions were computed using R version 4.1.2 via the package Survival version 3.5. A log-rank test was used to compare the survival distributions. Significance was set at *p* < 0.05.

## 3. Results

### 3.1. Complications and Reoperations

Donor site complications were observed in 5 of 39 patients (13%). Four patients had ankle valgus deformity; two patients with mild deformity were managed with an orthosis, while in the other two, the deformity was symptomatic and it was managed with syndesmotic screw fixation in one and varus osteotomy in the other one, 19 and 35 months after fibula harvesting, respectively. All these patients were skeletally immature at index surgery. The other patient had a first claw toe that was managed non-operatively.

During follow-up, 36 recipient site complications were observed in 26 of 39 patients (67%), after a mean of 21 months (0 to 107) from index surgery, requiring surgical revision in 19 patients (49%) after a mean of 31 months (0 to 107), while VFG removal was necessary in three patients (8%) after a mean of 86 months (25 to 190). Complications that required surgical revision were assessed in 12 of 26 (46%) tibial reconstructions after a mean of 28 months from index surgery (0 to 72), leading to VFG removal in one patient (4%), and in 7 of 13 (54%) femoral reconstructions after a mean of 36 months (1 to 107) from primary surgery, leading to VFG removal in 2 (15%) of them.

Four patients had postoperative common peroneal nerve palsy that completely recovered spontaneously in all cases; three were assessed in the tibial group and one in the femoral group.

Two patients had wound dehiscence within the first month after surgery, which was managed with surgical debridement and primary wound closure in one patient with a femoral reconstruction and with a medial gastrocnemius muscle flap in the other patient with tibial reconstruction; both reached wound healing. One patient with tibial reconstruction had screw-related pain, which resolved after screw removal 72 months after primary surgery.

Five nonunions were observed in four patients (10%) with tibial reconstructions, with hardware failure in one. None of the patients with femoral reconstruction experienced nonunion. Considering tibial reconstructions, nonunions occurred in 2 of 10 patients (20%) with bridging fixation and in 2 of 16 patients (12.5%) with non-bridging fixation. All nonunions were surgically managed with iliac crest autologous bone graft augmentation and new fixation, eventually reaching osteotomy healing in all cases, with a residual mild valgus deformity of the tibia in one patient.

In total, 14 patients (36%) had a fracture at a mean of 22 months (4 to 64) after primary surgery; fractures occurred in 8 of 26 patients (30%) with tibial reconstruction after a mean of 22 months (4 to 64) and in 6 of 13 patients (46%) with femoral reconstructions after a mean 21 months (1 to 43). Overall, 5 fractures were observed in 20 patients (25%) with bridging fixation, while 9 fractures occurred in 19 patients (47%) without bridging fixation. According to the reconstruction site, in the tibial group, 1 of 10 patients (10%) with bridging fixation and 7 of 16 patients (44%) with non-bridging fixation experienced a fracture; among femoral reconstructions, 4 of 10 patients (40%) with bridging fixation and 2 of 3 patients (67%) with non-bridging fixation had a fracture. All fractures involved the VFG except for one patient with femoral reconstruction, in which a non-displaced pertrochanteric fracture occurred after a fall. In five patients, four with tibial reconstruction, the fracture healed spontaneously with a non-operative treatment, using brace protection. In eight patients, fracture healing was obtained after new fixation with a bridging plate, with autograft augmentation in three cases. Finally, one patient with femoral reconstruction underwent to removal of non-viable VFG and reconstruction with a new intercalary allograft combined with the ipsilateral VFG, using a bridging fixation, with eventually healing.

Deep infection was observed in two patients (5%) with tibial reconstruction after 2 and 19 months from primary surgery, both healed after surgical debridement and IV antibiotics administration.

Five patients (13%) had local recurrence after a mean of 75 months (25 to 190) from tumor excision. Considering patients with tibial reconstruction, two had a local relapse in the soft tissue and were managed with surgical removal, leaving the VFG in situ, while in two patients, the local recurrence occurred in the bone; in one patient, the relapse occurred in the distal host bone and was managed with a new intercalary resection of the tibia preserving the previous VFG and a new ipsilateral VFG reconstruction, while in the other patient, the recurrence involved the soft tissue and the vessels and it was managed with a below knee amputation. Two of these patients were alive without evidence of disease at last clinical control at 81 and 270 months from primary surgery, while the other two died due to metastatic progression after 60 and 95 months follow-up. Considering femoral reconstructions, one patient had a proximal femur local recurrence 190 months after tumor excision, which was managed with proximal femur resection and reconstruction with modular prosthesis, retaining the distal part of the intercalary biologic reconstruction. None of the patients treated with intraepiphyseal resection experienced a local recurrence.

With the numbers available, neither age at surgery, skeletal maturity, gender, chemotherapy, resection length, or type of fixation influenced the risk of the above-mentioned complications in univariate and multivariate analysis.

In the last clinical control, a mean limb-length discrepancy of 2.8 cm (1 to 6) was observed in 21 patients (54%) and in 13 of 14 (93%) young patients who received an intra-epiphyseal resection. Limb-length discrepancies were managed with a shoe lift in all patients except two with 6 and 5 cm hypometria, one with tibial and the other with femoral reconstruction, in whom a lengthening with an external fixator and a lengthening nail, respectively, was performed.

### 3.2. Outcomes Score Function

At last clinical control, the mean MSTS score was 29.3 (23 to 30), with a mean score of 29 (23 to 30) in tibial reconstructions and 29 (27 to 30) in femoral reconstructions. Pain and emotional acceptance scored 5 points in all patients, function ranged between 4 and 5 points, while supports, walking, and gait ranged between 3 and 5 points. Regarding knee function, mean active flexion was 134° (100° to 140°), while an extension lag ranging between 5° to 20° was assessed in four patients. In patients with tibial reconstructions, mean ankle active flexion was 47° (20° to 50°) and mean active extension was 18° (5° to 20°). All patients with femoral reconstructions had a full hip range of motion.

### 3.3. Survivorship of Vascularized Fibular Grafts

The revision-free survival of the reconstructions, with revision surgery for any complication as the endpoint, was 59% (95% CI 45 to 76%) at 5 years and 50% (95% CI 36 to 69%) at 10 to 30 years (Figure 3). In tibial reconstructions, the revision-free survival of the reconstructions was 61% (95% CI 45 to 83%) at 5 years and 53% (95% CI 37 to 76%) at 10 to 30 years, while in femoral reconstructions, the revision-free survival of the reconstructions was 54% (95% CI 33 to 89%) at 5 years and 45% (95% CI 24 to 83%) at 10 to 30 years (Figure 4). All the complications that required surgical revision of the reconstruction, except five, occurred within the first five postoperative years. The overall survival of the reconstructions, with the removal of VFG as the failure endpoint, was 95% (95% CI 88 to 100%) at 5 to 15 years and 89% (95% CI 77 to 100%) at 20 to 30 years (Figure 5). Overall, three patients required VFG removal but only one due to mechanical complications.

## 4. Discussion

Osteosarcoma most frequently affects the metaphyseal region of the distal femur and proximal tibia and in most of the patients, a trans-physeal tumoral spread has been assessed, requiring an intra-articular resection to obtain safe surgical margins [1,3]. In around 20% of patients in whom the epiphyseal bone is not affected on MR images, an intercalary joint-sparing resection can be safely performed [4,5,6], preserving the native joint and ligament insertions. In young patients, according to their high functional expectations and potential long-life expectancy, the objective of reconstruction is to restore lower limb function with a low risk of reoperation and implant removal at long-term follow-up. VFG combined with massive allograft is one of the possible reconstructive options after intercalary resection around the knee. In the present study, we aimed to investigate the long-term results of this technique in patients treated for osteosarcoma around the knee with a joint-sparing resection, preserving less than 6 cm of meta-epiphyseal articular bone segment. Despite a consistent risk of mechanical complications in the first 5 postoperative years, more than 90% of the patients retained their primary reconstruction at the time of the most recent follow-up. Fracture risk is reduced using long-spanning plate fixation, while non-bridging fixation is suitable in intra-epiphyseal resection with scarce residual epiphyseal bone; in this case, weight-bearing forces may help to enhance union and hypertrophy of the fibula especially in young patients.

### 4.1. Limitations

First, this is a retrospective study with a relatively low number of patients due to the rarity of this tumor and the specific surgical reconstruction but, to our knowledge, this is one of the largest osteosarcoma series with this type of resection and reconstruction and such long-term outcomes. Second, there might have been a selection bias regarding the indication for intercalary joint-sparing resection rather than intra-articular resection but we usually aim to preserve native joints in all patients with primary bone tumors in whom the epiphyseal bone is not involved; in our experience, the functional advantages of preserving the native joints and their tendinous insertions are considerable. Third, five patients were not seen in the last 5 years; thus, they could have been treated for other complications elsewhere. Nonetheless, all these patients had more than 24 months of follow-up. Fourth, we did not analyze a control group with alternative intercalary reconstruction but we strongly believe in VFG reconstruction with allograft augmentation in patients undergoing juxta-articular joint-sparing resections leaving less than 3 cm of residual epiphyseal bone, in intra-epiphyseal resections, or in long intercalary resections; in these clinical situations, the allograft not only guarantees better mechanical stability but it also allows a more stable fixation; furthermore, it allows a biological tendinous reattachment of the patellar tendon in tibial resection above the tibial apophysis.

### 4.2. Complications and Reoperations

In our series, almost 50% of patients underwent surgical revision for complications. However, all but three of them retained their VFG at the last clinical control after a mean follow-up of 205 months.

Mechanical complications were the most frequently observed in our series. Fractures occurred in 36% of patients but all of them healed after new fixation or non-operative treatment, except for one patient in whom a non-viable VFG was removed, requiring a new reconstruction. One of the major drawbacks of the VFG reconstruction in the lower limb is the early low mechanical strength of the fibula; thus, the combination with a massive intercalary allograft is advocated [13]. Additionally, the type of fixation seemed to play an important role in the incidence of fractures [20,21]. We observed a lower fracture rate using a long bridging fixation either in tibial or femoral reconstructions. Thus, when feasible, a long-spanning plate fixation is recommended, adding a second plate to each osteotomy in very long resections, as reported by other authors [22]. Moreover, we assessed a higher incidence of fractures in femoral than tibial reconstructions, despite a higher proportion of spanning fixation and a similar resection length. Indeed, the femur is subject to higher mechanical stresses than the tibia and a double plate fixation should also be considered in this group of patients. Conversely, in intra-epiphyseal resections preserving a very short residual epiphyseal bone, a non-bridging fixation with metaphyseal screws and diaphyseal plate can be accepted, despite a higher risk of fractures. In this situation, the combination of VFG with massive allograft is advisable not only to improve mechanical stability but also to obtain a more stable fixation, where deficiency of consistent residual bone does not allow for a spanning plate fixation.

Nonunion was observed only in tibial reconstructions; this finding could be explained by the better soft tissue coverage of the femur compared to the tibia [23]. Furthermore, nonunion was more frequent in patients with bridging plate fixation, probably due to stress shielding that could inhibit fibular healing, remodeling, and hypertrophy. All nonunions healed after iliac crest autologous bone graft augmentation and new fixation.

The reconstruction with VFG after intercalary defects in the lower limb certainly has some drawbacks, such as the long surgical time related to the complexity of the fibula harvesting and anastomosis, which can be reduced through the simultaneous work of orthopedic and microsurgical teams, and the donor site morbidity, which, in our series, was 13%, requiring further surgeries only in two patients.

### 4.3. Outcomes Scores for Function

In our series, the mean MSTS score was 29.3 (23 to 30), with a mean score of 29 (23 to 30) in tibial reconstructions and of 29 (27 to 30) in femoral reconstructions and with excellent active motion of the hip, knee, and ankle. Such excellent functional results are probably related to the preservation of native joints and their tendinous and ligament insertions. Excellent functional results, in fact, were reported after joint-sparing intercalary reconstructions also in other studies [6,7,8,11,12,17,20,21]. Considering reconstructions with osteoarticular allografts around the knee, functional outcomes resulted poorer, in particular for distal femoral reconstruction [24], and even lower in modular prosthesis, with a mean MSTS of 76%, ranging between 30% and 100% [25].

### 4.4. Survivorship of Vascularized Fibular Grafts

The revision-free survival of the reconstructions was 59% (95% CI 45 to 76%) at 5 years and 50% (95% CI 36 to 69%) at 10 to 30 years; in tibial reconstructions, it was 61% (95% CI 45 to 83%) at 5 years and 53% (95% CI 37 to 76%) at 10 to 30 years, while in femoral reconstructions, it was 54% (95% CI 33 to 89%) at 5 years and 45% (95% CI 24 to 83%) at 10 to 30 years. In our series, despite the fact that almost 50% of the patients had a surgical revision for a complication, only three patients underwent VFG removal, with overall survival of the reconstructions of 95% (95% CI 88 to 100%) at 5 to 15 years and 89% (95% CI 77 to 100%) at 20 to 30 years.

Aponte-Tinao et al. [6], in a series of intercalary allografts around the knee after resection for osteosarcoma, reported an overall survival of the reconstruction of 68% at 10 years, with graft removal for failure in 26% of patients. The literature lacks studies focused on outcomes of intercalary reconstruction in patients treated for osteosarcomas but a graft survival of 71% and 80% at 10 years was reported for intercalary femoral prostheses and allografts, respectively [7], while a survival of 63% at 10 years was reported for intercalary tibial prostheses [8], with loosening as a major cause of failure in prosthetic reconstruction and fractures and infections in massive allograft.

Considering reconstructions in the lower limb with intercalary recycled autograft, complications requiring surgical revisions were reported in 13 of 20 patients (65%) treated for osteosarcoma in a series of 34 patients, with an overall graft survival of 91.2% at 10 years [11]. An attractive ad alternative option to VFG with allograft augmentation is represented by the combination of VFG and recycled autograft, with reported early bone union and a lower risk of complications [12]. Despite these favorable outcomes, intercalary frozen autografts are available only in selected patients without pathological fracture and aggressive osteolytic lesions and they are characterized by some drawbacks such as the absence of the specimen for histological examination of chemo-induced necrosis and surgical margins [11,12].

These reports, combined with our results, show that the augmentation of intercalary massive allograft with VFG, in order to merge the mechanical strength of the allograft with the long-term biological potential of VFG, seems effective to obtain a long-term durable reconstruction, in particular, in patients treated with chemotherapy. Certainly, this technique represents a challenging option requiring a well-trained microsurgical team but we believe the biological potential of a VFG constitutes a major advantage.

In our series, 54% of all patients and 93% of the skeletally immature patients treated with an intra-epiphyseal resection had a limb-length discrepancy that was surgically managed in two cases; distal femoral and proximal tibial epiphysis contributed to lower limb growth to 35% and 30%, respectively [26]. Expandable prostheses have been introduced to overcome this problem; nonetheless, a very high rate of prosthetic revision has been reported, in particular for distal femoral and proximal tibial prostheses, with a failure-free survival lower than 40% at 10 years and lower than 20% at 20 years [27].

Finally, we had a slightly higher incidence of local recurrences compared to other reports, with a similar overall survival [6,17,25]. Most of our patients with local relapse retained their VFG, except one patient treated with below-knee amputation and another one who received a proximal femoral resection for local recurrence 16 years after primary surgery. In this case of very late recurrence, a new primary osteosarcoma can be postulated. None of our patients who underwent intraepiphyseal resection experienced a local recurrence, confirming that the growth plate represents an effective barrier to tumoral extension [4].

## 5. Conclusions

VFG represents an effective reconstructive option after joint-sparing intercalary resection around the knee for osteosarcoma; it can be indicated for young and active patients with long life expectancy. A long-spanning plate fixation is recommended to decrease the risk of fracture, frequently observed during the first postoperative years, while non-bridging fixation represents an option after intra-epiphyseal resection when the short residual epiphyseal bone does not allow a stable spanning fixation. In our experience, although surgical revision was performed in almost half of the patients, more than 90% were able to retain their primary reconstruction at the last clinical follow-up. Most of the failures were fractures and nonunions that occurred within the first postoperative years; once the union of the graft was achieved, the complication rate was very low, confirming that a biological reconstruction could also result in a long-lasting solution in patients treated with chemotherapy.

In conclusion, we believe that VFG combined with allograft is an effective option to reconstruct a functional lower extremity after intercalary resection around the knee for osteosarcoma, providing a long-lasting durable reconstruction with excellent functional results.

## Figures and Tables

**Figure 1 cancers-16-01672-f001:**
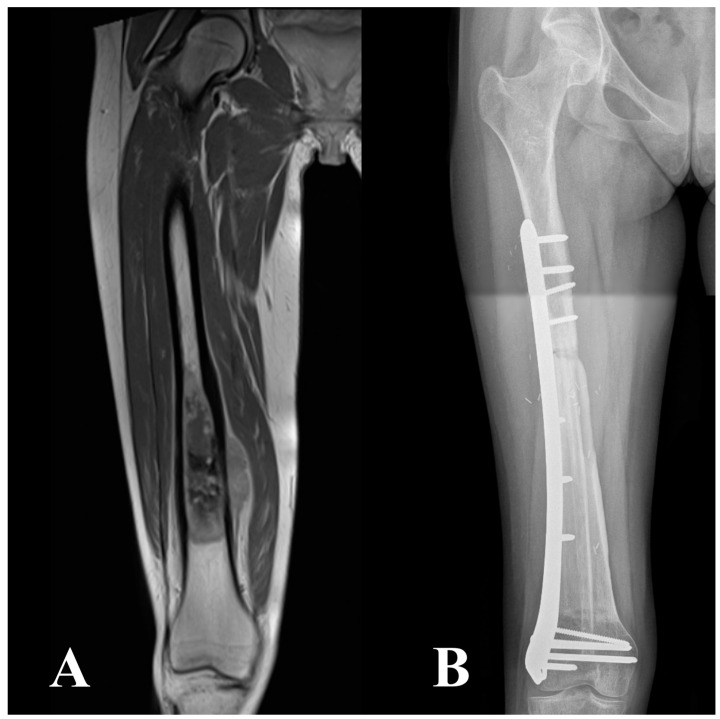
Distal femur meta-diaphyseal high-grade osteosarcoma on T1-weighted MRI (**A**). The patient underwent an intercalary resection and reconstruction with VFG combined with massive allograft, using a long spanning fixation (**B**).

**Figure 2 cancers-16-01672-f002:**
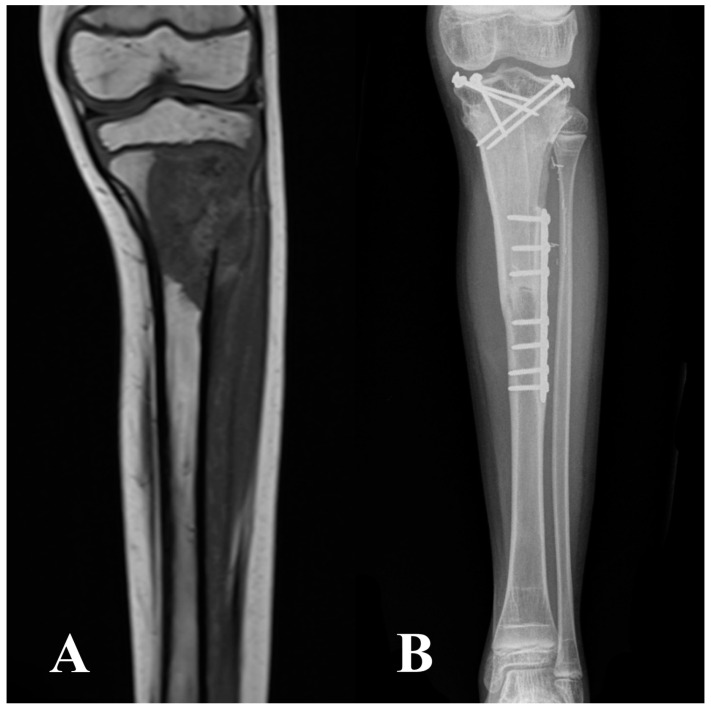
Proximal tibia metaphyseal high-grade osteosarcoma on T1-weighted MRI (**A**). The patient underwent an intraepiphyseal resection and reconstruction with VFG combined with a massive allograft, using a non-spanning fixation with metaphyseal screws and diaphyseal plate (**B**).

**Figure 3 cancers-16-01672-f003:**
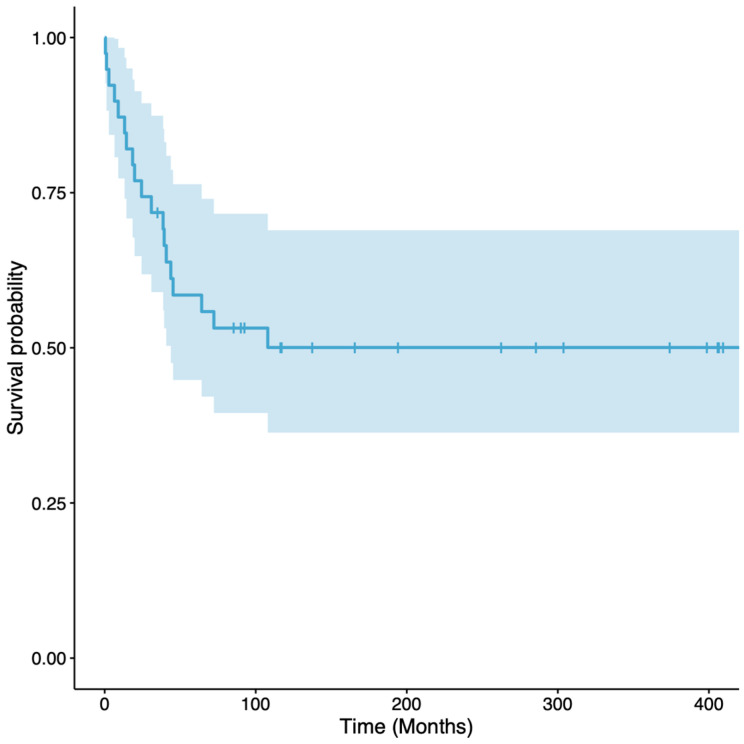
Graph showing revision-free survival, with surgical revision for any complication as the endpoint. The survival was 59% (95% CI 45 to 76%) at 5 years and 50% (95% CI 36 to 69%) at 10 to 30 years. The light blue area represents the CIs.

**Figure 4 cancers-16-01672-f004:**
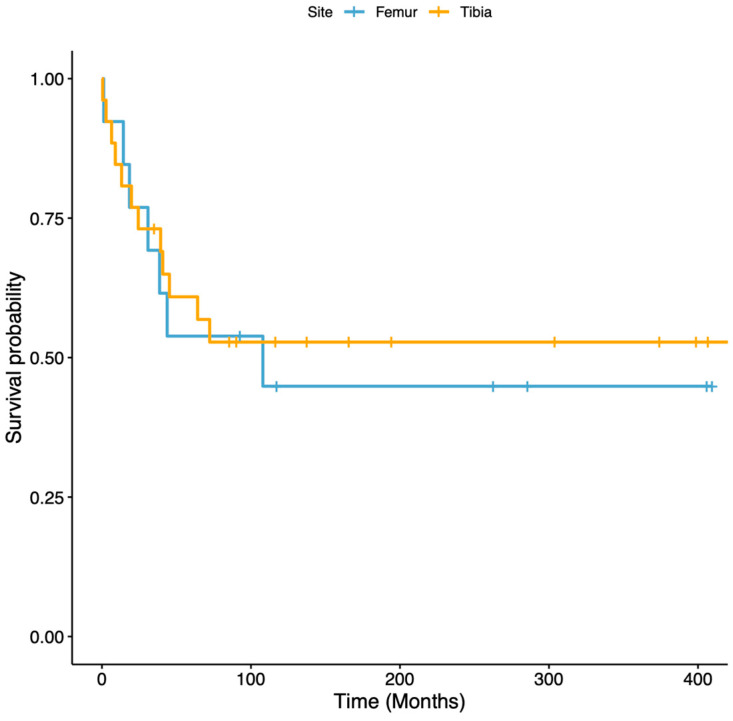
Graph showing revision-free survival, with surgical revision for any complication as the endpoint of femoral (blue line) and tibial (orange line) reconstructions. No statistical differences were assessed between these two survivals.

**Figure 5 cancers-16-01672-f005:**
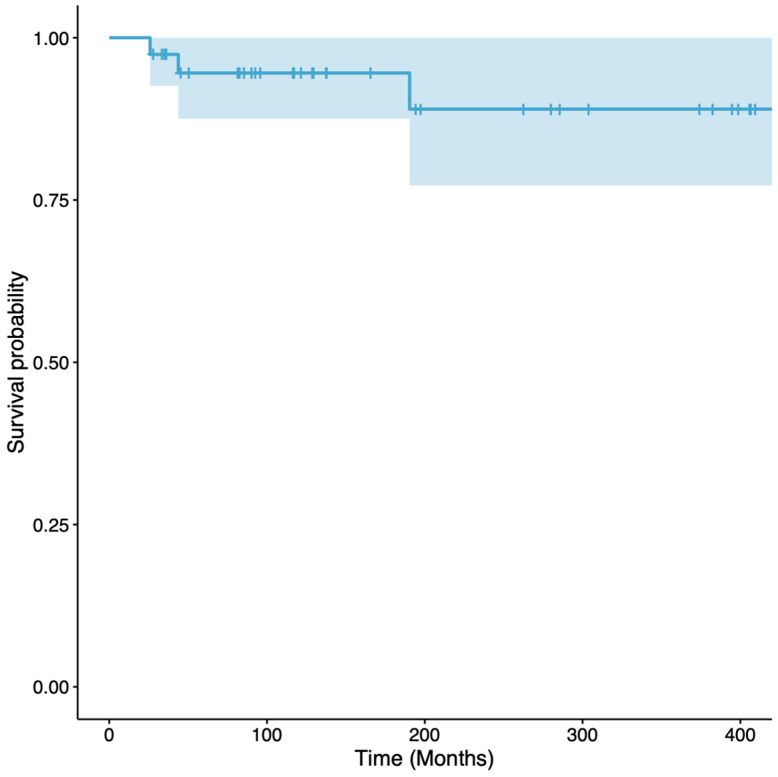
Graph showing VFG removal-free survival, with graft removal for any complication as the endpoint. The survival was 95% (95% CI 88 to 100%) at 5 to 15 years and 89% (95% CI 77 to 100%) at 20 to 30 years The light blue area represents the CIs.

**Table 1 cancers-16-01672-t001:** Patient’s characteristics.

Patients’ Characteristics	%/Average (Number/Range)	
	Femur	Tibia
Men	77 (10)	65 (17)
Women	23 (3)	35 (9)
Age (years)	15 (7–37)	17 (7–67)
Osteosarcoma grading		
High-grade	100 (13)	81 (21)
Low-grade	0 (0)	9 (5)
Type of resection		
Intercalary	77 (10)	58 (15)
Intraepiphyseal	23 (3)	42 (11)
Resection length (cm)	15 (10–25)	14 (10–20)
Proximal residual juxta-articular bone	16.5 (6–26)	3 (1–6)
Distal residual juxta-articular bone	3.8 (2–6)	16 (7–21)
Fibular resection length	17.6 (12–28)	17 (13–23)
Fixation method		
Single bridging plate	77 (10)	35 (9)
Double bridging plate	0 (0)	4 (1)
Metaphyseal screws + diaphyseal plate	15 (2)	42 (11)
Only screws	8 (1)	12 (3)
Proximal and distal plates	0 (0)	8 (2)

## Data Availability

Dataset available on request from the authors.

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
