# Peer review of "Joint-Sparing Resection around the Knee for Osteosarcoma: Long-Term Outcomes of Biologic Reconstruction with Vascularized Fibula Graft Combined with Massive Allograft"

_cancers, 2024, doi:10.3390/cancers16091672_

Round 1
Reviewer 1 Report
Comments and Suggestions for Authors
Roberto Scanferla and co-authors present a high quality and well-written experimental manuscript focused on joint-sparing resection around the knee for osteosarcoma with respect to long-term outcomes of biologic reconstruction with vascularized fibula graft combined with massive allograft.
Authors aimed to address following questions: What was the complication rate of vascularized fibula graft (VFG) combined with massive allograft in patients treated with joint-sparing resection around the knee for a high-grade osteosarcoma? What was the long-term survivorship of VFG free from revision and graft removal? What were the functional results as assessed by the Musculoskeletal Tumor Society (MSTS) score?
Authors treated 39 patients treated in their Unit for an osteosarcoma around the knee with intercalary resection and reconstruction with VFG combined with massive allograft were included; 26 patients underwent to intercalary tibial resection, while 13 underwent to intercalary femoral resection.
Authors found that the mean follow-up was 205 months (28 to 424). Complications that required surgery were assessed in requiring surgical revision in 19 patients (49%) after a mean of 31 months (0 to 107), while VFG removal was necessary in three patients (8%). The revision-free survival of the reconstructions was 59% at 5 years and 50% at 10 to 30 years. The overall survival of the reconstructions was 95% at 5 to 15 years, and 89% at 20 to 30 years. The mean MSTS score was 29.3 (23 to 30).
Finally, authors conclude that VFG represents an effective reconstructive option after joint-sparing intercalary resection around the knee for an osteosarcoma.
Overall, the manuscript is highly valuable for the scientific community and should be accepted for publication.
======================
Other comments to authors:
1) Please check for typos throughout the manuscript.
2) Please improve figures/tables where appropriate.
3) With regards to solid tumors - authors are kindly encouraged to cite the following article that describes novel approaches for treatment of solid tumors, which can be relevant for sarcomas. DOI: 10.3390/biomedicines11020626
Author Response
Thank you for appreciating our work.
1) We checked the typos (line 183 and 403) and corrected them.
2) We checked tables and figures, but we didn't find any aspect to improve, in our opinion. If you have any suggestions, please give them to us.
3) We read the article (very interesting and well written), but it is not strictly related to the aims of our work, thus this reference is probably unnecessary.
Once again, thank you for your interest in our work and for your suggestions.
Reviewer 2 Report
Comments and Suggestions for Authors
This paper deal about long term outcome and complication rates in patients with tibial and femoral osteosarcoma treated with VFG. This data are lacking in the literature. The study is retrospective but methodologically correct and is conducted with evident intellectual honesty as demonstrated by the real occurrence of complications. Paper is well written resulting easily readable and understandable The study may contribute to sustain the use of this reconstructive technic and my address future prospective studies in the field.
I only suggest to remove from figure legend "this figure" and substitute with images showing.. or graph showing.. In line 183 ll need to be corrected in All.
Comments on the Quality of English LanguageEnglish language is substantially good resulting easily readable and understandable
Author Response
Thank you for appreciating our work.
We corrected the figures legend according to your suggestions and the typos on line 183.
Once again, thank you for your interest in our work